# Regulation of EWSR1-FLI1 Function by Post-Transcriptional and Post-Translational Modifications

**DOI:** 10.3390/cancers15020382

**Published:** 2023-01-06

**Authors:** Le Yu, Ian J. Davis, Pengda Liu

**Affiliations:** 1Lineberger Comprehensive Cancer Center, The University of North Carolina at Chapel Hill, Chapel Hill, NC 27599, USA; 2Department of Biochemistry and Biophysics, The University of North Carolina at Chapel Hill, Chapel Hill, NC 27599, USA; 3Department of Genetics, The University of North Carolina at Chapel Hill, Chapel Hill, NC 27599, USA; 4Department of Pediatrics, The University of North Carolina at Chapel Hill, Chapel Hill, NC 27599, USA

**Keywords:** Ewing sarcoma, EWSR1-FLI1, transcriptional regulation, post-translational modifications, targeted therapy

## Abstract

**Simple Summary:**

Ewing sarcoma is a malignant pediatric bone cancer currently lacking targeted therapy. In the US there are ~200 patients diagnosed each year and relapse is associated with resistance to the standard-of-care chemotherapy. Thus, it remains an urgent unmet medical need to develop effective new cures for Ewing sarcoma. It is well-characterized that Ewing sarcoma is largely driven by unique gene fusions, with EWSR1-FLI1 being the most prevalent. In this review, we summarize up-to-date regulatory mechanisms for the onco-fusion protein EWSR1-FLI1 in Ewing sarcoma, including both post-transcriptional and post-translational modifications, to reveal knowledge gaps and propose potential new therapeutic directions.

**Abstract:**

Ewing sarcoma is the second most common bone tumor in childhood and adolescence. Currently, first-line therapy includes multidrug chemotherapy with surgery and/or radiation. Although most patients initially respond to chemotherapy, recurrent tumors become treatment refractory. Pathologically, Ewing sarcoma consists of small round basophilic cells with prominent nuclei marked by expression of surface protein CD99. Genetically, Ewing sarcoma is driven by a fusion oncoprotein that results from one of a small number of chromosomal translocations composed of a FET gene and a gene encoding an ETS family transcription factor, with ~85% of tumors expressing the EWSR1::FLI1 fusion. EWSR1::FLI1 regulates transcription, splicing, genome instability and other cellular functions. Although a tumor-specific target, EWSR1::FLI1-targeted therapy has yet to be developed, largely due to insufficient understanding of EWSR1::FLI1 upstream and downstream signaling, and the challenges in targeting transcription factors with small molecules. In this review, we summarize the contemporary molecular understanding of Ewing sarcoma, and the post-transcriptional and post-translational regulatory mechanisms that control EWSR1::FLI1 function.

## 1. Introduction

Ewing sarcoma, first described in 1921 by James Ewing, encompasses three tumor types: “classic” Ewing sarcoma of bone, malignant small cell tumors of the chest wall (Askin’s tumor), and primitive neuroectodermal tumors of soft tissue origin (PNET) [1]. Although the definitive cell of origin remains unknown, Ewing sarcoma is thought to originate from mesenchymal progenitor cells in bone and soft tissue [2]. Ewing sarcoma, the second most common pediatric bone malignancy, constitutes 10% to 15% of all bone sarcomas and occurs most commonly in children and young adults [3]. The median age of diagnosis of this disease is 14–15 years [4,5]. Ewing sarcoma mostly occurs in the pelvis, mid-shaft bones and femur; however, it can occur in all bones and many soft tissues. Ewing sarcoma is considered as a systemic disease ab initio with the lung being the most common site of metastasis site. Usually, pain and swelling are observed at disease sites. Over the past century, our understanding of the development, progression and treatment of Ewing sarcoma has advanced greatly. Important discoveries include the identification of the molecular architecture of Ewing sarcoma, defining Ewing sarcoma pathology, and the development of successful therapies to treat Ewing sarcoma.

Ewing sarcoma is virtually uniformly associated with a gene fusion composed of the N-terminus of an RNA-binding protein, mostly commonly EWSR1 (ES breakpoint region 1) and the carboxyl terminal DNA-binding domain of an ETS (erythroblast transformation-specific) family transcription factor (Table 1): (1) ~85% cases exhibit a t(11;22)(q24;q12) translocation, which joins EWSR1 on chromosome 22 with FLI1 (friend leukemia integration 1) on chromosome 11, resulting in the EWSR1::FLI1 onco-fusion gene; (2) ~10% cases bear a t(21;12)(q22;q12) translocation, generating an EWSR1::ERG fusion gene; (3) and the remaining 5% cases contain other EWSR1-ETS fusion genes such as EWSR1::FEV, EWSR1::ETV1, EWSR1::ETV4 and others [6,7,8,9,10,11]. FUS and TAF15, other TAF family RNA-binding proteins, are also involved in the fusions with ETS proteins in a small percentage of Ewing sarcoma [12] (Table 1). Moreover, a reciprocal chromosomal translocation resulting in FLI1-EWSR1 fusion was also observed and reported in Ewing sarcoma cell lines and tumors [13]. The exact origin(s) for Ewing sarcoma remains unclear although mesenchymal stem cells [14], neural stem cells [15] and osteochondrogenic progenitors [16] have been proposed as candidates. To date there is no genetic murine model to mimic human Ewing sarcoma.

In this review, we summarize the up-to-date understanding of the pathology and molecular features for Ewing sarcoma, and various EWSR1::FLI1 regulatory mechanisms especially post-translational modifications, aiming to provide new insights for identifying novel drug targets to fight against this deadly pediatric cancer.

## 2. The Pathology of Ewing Sarcoma

Ewing sarcoma is not considered a familial cancer, although a genetic predisposition has been identified in population level studies [17]. Histologically, the tumors consist of generally uniform round cells with vesicular nuclei of finely dispersed chromatin and hyaline cytoplasm [18].More than 95% of tumors express the cell surface protein CD99 (also named MIC2), which has been used as a marker for Ewing sarcoma [19]. However, CD99 expression is not specific for Ewing sarcoma. CD99 is also expressed in certain normal tissues and other mesenchymal tumors. However, negative CD99 immunogenicity strongly argues against the diagnosis of Ewing sarcoma. FLI1 staining in nuclei is more specific for clinically defining Ewing sarcoma [20]. However, IHC is not sufficient for the diagnosis of Ewing’s sarcoma, and molecular translocation testing is required to exclude other round-cell sarcomas, per the guidelines from the 2020 WHO classification [21]. Other markers such as neuron-specific enolase (NSE), S-100 protein, CD57, neurofilaments, cytokeratin, desmin, caveolin-1, NK2 homeobox 2 (Nkx-2.2) or immunohistochemical markers such as B-cell CLL/lymphoma 11B (BCL11B) and Golgi glycoprotein 1 (GLG1) have been investigated in Ewing sarcoma diagnosis, especially in cases with negative CD99 detection [22,23,24,25,26,27,28]. In addition, alterations in DNA methylation have also been observed to distinguish Ewing sarcoma [29]. Together, currently a commonly accepted standard for the diagnosis of Ewing sarcoma includes a consistent histological morphology, staining for CD99, and evidence of EWSR1 rearrangement by fluorescence in situ hybridization, PCR or sequencing.

## 3. Therapies in Ewing Sarcoma

For nearly 40 years, the treatment for Ewing sarcoma has included systemic chemotherapy and local treatments, including radiotherapy and surgery. If surgery is not feasible or highly morbid, then radiation therapy may be an exclusive treatment. Radiotherapy can be combined with surgery when adequate surgical margins are anticipated to be difficult to achieve or margins are found to be positive following resection. Doxorubicin, vincristine, cyclophosphamide, etoposide and ifosfamide are the standard of care for patients in the US. Although the 5-year survival rate has been improved from less than 20% to greater than 70%, the recurrence rate remains high, with some patients relapsing many years from the end of treatment [30]. For relapsed patients, survival rates are less than 30%. Surgery (uncommon except for local recurrences), radiotherapy and chemotherapy (irinotecan and temozolomide) are used for the treatment with survival rates of up to 50% in selected patients [31,32]. Thus, there remains an urgent medical need to develop new treatments for Ewing sarcoma. Targeted treatments require a better understanding of the molecular mechanisms driving the development of Ewing sarcoma and chemo-resistance.

Although the identification of a tumor-specific oncogenic fusion on which Ewing sarcoma is dependent points to the perfect drug target, at present there is not an FDA-approved therapeutic. This is partially due to the challenges in directly targeting EWSR1::FLI1 [33,34]. YK-4-279 was identified as an interactor of RNA helicase A, which interacts with EWSR1-FLI1 [35], although a recent study suggests that TK216, the related molecule in clinical testing, targets microtubules [36]. Histone deacetylase inhibitors have also been shown to affect Ewing sarcoma cell proliferation possibly through EWSR1::FLI1-mediated transcription and chromatin regulation, although they have yet to demonstrate a clear signal in clinical testing [37,38,39,40,41,42,43,44,45]. Other therapies in clinical testing include an LSD1 inhibitor, that has been shown to reverse the EWSR1::FLI1 transcriptional program [46]. LSD1 (lysine specific demethylase 1) is a protein lysine demethylase first identified to demethylate H3K4me1/2 [47] and later shown to demethylate non-histone proteins [48]. EWSR1::FLI1 recruits LSD1 to NuRD (nucleosome remodeling and histone deacetylase) complexes to suppress transcription of genes including LOX and TGFBR2 [49]. Phase I trials have demonstrated efficacy of the combination of the PARP inhibitor talazoparib with temozolomide [50]. Other mechanisms for potential therapeutic targeting include competition of EWSR1::FLI1 with MRTFB on binding to gene promoters leading to the suppression of MRTFB-mediated transcription of TAZ (also known as transcriptional coactivator with PDZ-binding motif) [34]. As a downstream effector of the Hippo signaling pathway, TAZ has been shown to facilitate cell migration through multiple mechanisms including limiting cytoskeletal and focal adhesion maturation [51], altering metabolic programs [52] and others. Although EWSR1::FLI1 is indispensable to maintain Ewing sarcoma growth, increased expression of EWSR1::FLI1 is also not tolerated as it causes cell growth arrest and cell death [53]. Thus, EWSR1::FLI1 protein homeostasis is tightly controlled, and variability in EWSR1-FLI1 might enhance metastatic potential [54].

## 4. Oncogenic Mechanisms of EWSR1-FLI1 in Ewing Sarcoma

EWSR1::FLI1 is necessary to maintain Ewing sarcoma proliferation and survival and exerts an ability to transform human primary mesenchymal stem cells [55,56]. There are two common types of EWSR1::FLI1 fusions in patients: type 1 (fused by EWSR1 exon 7 with FLI1 exon 6) and type 2 (fused by EWSR1 exon 7 with FLI1 exon 5) fusions. EWSR1 contributes an unstructured domain that engages in phase separation and modulates transcriptional control and RNA splicing through interactions with the SWI/SNF complex and HNRNPs [57,58]. Both type 1 and type 2 EWSR1::FLI1 fusions retain the C-terminal DNA-binding domain of FLI1, thus serving as novel aberrant transcription activators. EWSR1::FLI1 binds to microsatellite regions that contain repeats of GGAA, the core of the ETS DNA recognition elements [59,60,61]. EWSR1::FLI1 binding promotes chromatin accessibility, neo-enhancer development and transcription regulation [60]. In addition, EWSR1::FLI1 also directly binds enhancer elements to modulate gene expression [62]. Overall, the chimeric EWSR1::FLI1 transcription factor promotes malignant transformation by regulating the transcription of a large number of downstream target genes [63].

Multiple studies have explored which EWSR1::FL1 transcriptional targets account for its transformation capacity and may serve as possible viable drug targets. Early studies recovered 99 putative transcription factors co-immunoprecipitated with EWSR1::FLI1-bound chromatin. *MK-STYX* (a MAPK phosphatase-like protein) identified by this approach was further validated [64] that in part mediates the oncogenic properties of EWSR1-FLI1. Moreover, a microarray analysis in Ewing sarcoma A673 cells revealed that depletion of endogenous EWSR1::FLI1 by retroviral siRNAs upregulated 320 genes and downregulated 1151 genes, among which *NKX2.2* was reported as a critical EWSR1-FLI1 downstream target [65]. A meta-analysis further defined a “core” EWSR1-FLI1 transcriptional signature [66] by integrating transcriptional profiling data from distinct cell line models including NIH-3T3 [67,68], primary human fibroblasts [69], primary bone marrow-derived mesenchymal progenitor cells [55], mesenchymal stem cells [70], rhabdomyosarcoma cells [71], neuroblastomas [72], patient-derived Ewing sarcoma cell lines [73] and others. These EWSR1::FLI1 targets include transcription factors such as NKX2.2, GLI1, FOXM1, DAX-1, secreted proteins such as cholecystokinin and LOX, neural crest developmental genes such as MAPT [71], cell cycle regulators such as p21 [74], as well as kinases such as PIM3 [68], AURKA and AURKB [75,76,77,78]. However, EWSR1::FLI1 target identification outside of the context of cell-of-origin must be interpreted with caution. Recently, neo-transcripts have also been identified from silent genome regions uniquely activated by the EWSR1::FLI1 fusion, suggesting these neo-genes might be targetable for Ewing sarcoma treatment [79].

In addition to protein-coding genes, the long noncoding RNA *EWSAT1* was found as an EWSR1::FLI1-induced product by RNA-seq analysis using primary pediatric human mesenchymal progenitor cells [80]. EWSR1::FLI1 repressed *miR-708* expression to indirectly induce *EYA3* transcription [81], and inhibited expression of the tumor suppressive *miR-145* [82,83], and other miRNAs including *miR-22*, *miR29a*, *miR-100*, *miR-125b*, *miR-221/222* and *miR-271* [84] to modulate tumor growth.

Moreover, EWSR1::FLI1 not only directly modulates the transcription, but also controls transcript degradation [85] and alternative splicing [86] as additional regulatory mechanisms to govern RNA abundance.

In addition to directly modulating chromatin conformation and gene expression, EWSR1::FLI1 also induces genome instability that facilitates tumorigenesis. EWSR1::FLI1 promotes transcription to cause R-loops, which titrate BRCA1 away from sensing damaged DNA thus blocking BRCA1-mediated DNA damage repair [87]. Together, EWSR1::FLI1 utilizes at least three distinct mechanisms to promote Ewing sarcoma growth (Figure 1).

The EWSR1::FLI1 is located in granules in nuclei [88]. An unstructured domain in EWSR1 mediates the phase transition of EWSR1::FLI1 and enhancer activation [89]. Similarly, the low-complexity domain interactions among EWSR1::FLI1 are also reported (which are transient and dynamic) to promote transcriptional activity at a narrow optimal level [90]. Increasing concentrations of EWSR1-FLI1 low-complexity domain interactions promotes EWSR1::FLI1 phase transition in the nucleolus and suppresses the EWSR1::FLI1 transcriptional [91]. Thus, depending on the levels of EWSR1::FLI1 low-complexity domain interactions, phase transition can either promote or suppress transcription.

## 5. EWSR1-FLI1 Regulatory Mechanisms

### 5.1. Transcriptional Regulation

Transcriptional activation of the EWSR1::FLI1 fusion gene was accompanied by deposition of histone markers on the EWSR1 promoter [92]. Analysis of clinical cases by conventional cytogenetics, fluorescence in situ hybridization and nested PCR revealed that H3K4me3, H3K9ac and H3K27ac were significantly enriched in the EWSR1 promoter in Ewing sarcoma to facilitate transcription [93,94].The transcription factor SP1 directly binds to the EWSR1::FLI1 promoter to trigger transcription, a process induced by activated PI3K/Akt signaling [95]. In addition, hypoxia promotes *EWSR1::FLI1* transcription in a HIF-1a-dependent manner [96].

The RNA-binding protein HNRNPH1 was shown to facilitate Ewing sarcoma cells to properly express EWSR1 exon 8 genomic breakpoint fusions [58,97]. The stability of *EWSR1::FLI1* mRNA in approximately 10% of Ewing sarcomas is regulated by the carcinoembryonic RNA-binding protein LIN28B. Deletion of LIN28B led to decreased EWSR1::FLI1 expression, which affects the self-renewal and tumorigenicity of Ewing sarcoma cells [98]. Knocking down *CRM1* (XPO1) significantly inhibited the expression of EWSR1::FLI1 fusion proteins at the post-transcriptional level with unknown mechanism(s) [99]. On the other hand, *miR-145* was shown to suppress EWSR1::FLI1 transcription [82]. A small-molecule screen identified that histone deacetylase inhibitors decrease EWSR1::FLI1 levels, possibly contributing to the cytotoxic effect of these drugs on cells [40,45].

### 5.2. Translational Regulation

The stability of de novo synthesized EWSR1-FLI1 proteins can be reduced by treatment with lovastatin or tunicamycin, leading to reduced protein levels and decreased Ewing sarcoma cell growth [1,2,100]. In addition to suppressing EWSR1::FLI1 protein synthesis, tunicamycin also suppressed N-linked glycosylation (likely N-linked glycosylation of IGF1-R) that suppresses function and cell growth [101]. Notably, although EWSR1::FLI1 expression contributes to a proliferative phenotype, reduced levels of EWSR1::FLI1 proteins have been shown to decrease proliferation but induce a more motile phenotype [33,34].

### 5.3. Protein-Level Regulation

EWSR1::FLI1 is also regulated post-translationally (Figure 2). This includes regulation of the physical properties of EWSR1-FLI1 protein by various post-translational modifications that regulate protein function in an acute and spatial manner, as well as various binding proteins that either facilitate or suppress EWSR1::FLI1 function on chromatin.

Post-translational control: Multiple post-translational modifications of EWSR1::FLI1 have been described that regulate transcriptional activity and function in both tempo- and spatial manners (Figure 2). Specifically, EWSR1::FLI1 is phosphorylated at Thr79 in the N-terminal EWSR1 domain upon DNA damage or mitogen stimulations by ERK1/ERK2/JNK or p38-MAPKs, respectively. This modification presumably stimulates dimer formation and transcriptional activity [102]. O-GlcNAcylation of EWSR1::FLI1 was also reported to positively regulate oncogenic function in Ewing sarcoma [103]. In addition, acetylation of the C-terminal FLI1 region by PCAF increases its DNA-binding ability to potentiate transcriptional activity [104]. Ubiquitination of EWSR1::FLI1 is observed on the Lys380 residue that primes the protein for proteasomal degradation [105]. In addition, lysosome-dependent protein degradation is also reported [106]. Multiple E3 ubiquitin ligases have been observed to mediate EWSR1::FLI1 ubiquitination and degradation in Ewing sarcoma, including TRIM8 [53] and SPOP [107]. Deubiquitinases, including USP19 [108] and OTUD7A [107], stabilize EWSR1-FLI1 protein. Notably, CK1-mediated Ser486/487/488 (based on type-II variant) phosphorylation primed EWSR1::FLI1 for recognition and regulation by either SPOP or OTUD7A [107]. Recently, we identified 7Ai, a putative small-molecule OTUD7A inhibitor that suppresses EWSR1::FLI1 protein expression and subsequent Ewing sarcoma cell and tumor growth [107]. Notably, it is currently largely unclear if these modifications are also present in the native proteins.

Binding proteins: Control of EWSR1::FLI1 transcriptional activity is also achieved by multiple protein interactors. The RNA helicase A (RHA) binds EWSR1::FL1 to enhance its transcriptional activity [109]. PARP-1 also interacts with EWSR1::FLI1 to facilitate transcription [110]. hsRBP7, as a subunit of RNA polymerase holozyme II (Pol II) interacts with EWSR1::FLI1 through its EWSR1 portion [111]. EWSR1::FLI1 also complexes with EWSR1 (and with RNA Pol II) to exert its transcriptional activity. EWSR1::FLI1 forms a ternary complex with ELK1-SAP1a to bind SRF using the unique R-domain near the FLI1 DNA-binding region to upregulate ERG1 expression [112]. BARD1 as a putative tumor suppressor, interacts with the N-terminus of EWSR1::FLI1 [113]. The FOS-JUN dimer also interacts with EWSR1::FLI1 to bind AP1 sequences [114]. Additionally, EWSR1::FLI1 recruits the BAF complex to tumor-specific enhancers to promote activation of target genes [89]. Steroid-dependent translocation of EWSR1::FLI1 and glucocorticoid receptor into nuclei leads to EWSR1::FLI1 binding to the glucocorticoid receptor to enhance glucocorticoid receptor-mediated oncogenic transcription to facilitate Ewing sarcoma growth and migration [115]. Proteomics analysis also reveals that CIMPR (cation-independent mannose 6-phosphate receptor) as a EWSR1::FLI1 binding partner that regulates EWSR1::FLI1 degradation in a lysosome-dependent pathway [106]. Although these binding proteins exert important but distinct roles in Ewing sarcoma, whether they participate in different protein sub-complexes at various cellular compartments or under distinct pathophysiological functions remains to be further determined. Notably, to date there is no genetic mouse model to study Ewing sarcoma pathology, biology and the testing of therapeutics’ effects although zebrafish [116] and drosophila [117] models have been developed. There are limited number of established PDX murine models using both subcutaneous and tibial implantation.

## 6. Concluding Remarks and Future Perspectives

Chromosomal translocations between chr11 and chr22 occur specifically in Ewing sarcoma to generate the fusion oncogene EWSR1::FLI1 [118,119,120]. The exact mechanism resulting in this chromosomal translocation remains elusive. Although EWSR1::FLI1 presents a unique therapeutic target in Ewing sarcoma, to date no effective targeted therapies have been validated and approved to treat Ewing sarcoma. The first-line therapy in clinic relies on intensive chemotherapy combined with surgery and radiation. Presently, there is no standard of care for chemo-resistant, relapsed patients.

Firstly, more efforts in the molecular classification of Ewing sarcoma tumor subtypes would benefit the discovery of vulnerabilities and development of effective cures. Loss of *TP53* and *STAG2* is associated with a poor outcome [121]. Unlike breast cancer that is classified into basal-like, luminal and other types of tumors based on the genetic architecture, and prostate cancer is divided into *AR^+^/AR*^−^ or castration-resistant subtypes, treatment relevant molecular subtypes have yet to be identified. This effort may explain the heterogeneity observed in Ewing sarcoma [122] and may help further direct proper treatments/combination treatments for subtypes of Ewing sarcoma patients to improve treatment efficacy. However, the rarity of Ewing sarcoma limits the availability of a large cohort of patient samples that can be used to faithfully perform molecular subtyping. In addition, how to incorporate this information into clinical designs remains as another challenge. Notably, efforts in genotyping Ewing sarcomas [123,124,125,126] have begun to shed new light onto the Ewing sarcoma pathology.

Secondly, establishing genetic models that recapitulate Ewing sarcoma initiation and progression would help identify those key molecular events necessary for tumor development—although the unique positional relationship between GGAA-containing microsatellite enhancers and relevant genes unique to the human genome may preclude this approach. Although EWSR1::FLI1 is the driver for Ewing sarcoma, its expression in nontumor cells induces apoptosis and fails to promote tumor formation, bringing challenges in establishing genetic models and suggesting the relevance of additional genetic alternations and cellular context [127,128]. Given that Ewing sarcoma zebrafish models require the deletion of p53 or expression of anti-apoptotic BCL-2 family proteins [129], additional genetic manipulations may be necessary to facilitate EWSR1::FLI1-driven mouse models. Establishing a valid genetic murine model will greatly facilitate the understanding of key steps in Ewing sarcoma initiation and development, which in turn will facilitate new biomarker identification and drug target discovery.

Thirdly, developing additional patient-derived xenografts (PDX), as well as immune-competent murine models would benefit preclinical studies. To date, validation of effects of gene function or treatment effects in Ewing sarcoma largely relies on in vitro cell-line studies and xenografted mouse models in immunodeficient settings [130,131]. PDX mouse models mimic aspects of cancer development [132]. A limited number of Ewing sarcoma PDX models (https://www.pdxfinder.org/, accessed on 1 November 2022) have been established, and both flank transplantation and orthotopic transplantation (or -patient-derived orthotopic xenograft (PDOX)) have been successfully developed [131,133,134]. The use of PDX and PDOX models allows for optimized conditions for drug development and precise cancer therapy. In addition, establishing Ewing sarcoma cell lines derived from patients that can survive in immune-competent mice would greatly facilitate the studies using immune-checkpoint blockades and CAR-T cell therapies in Ewing sarcoma.

Lastly, understanding drug resistance mechanisms would provide new avenues for treatment modality. Efforts have been devoted to deciphering critical downstream transcriptional targets of EWSR1::FLI1, its binding partners, modifications and roles in regulating Ewing sarcoma proliferation and motility. Given Ewing sarcoma is a systemic disease, biomarkers that predict response to chemotherapy would be relevant. Although ~50% patients respond to chemotherapy initially, once relapsed, most patients are insensitive to chemotherapy. In addition, frequent intensive chemotherapy decreases patients’ quality of life. How to identify patients who are likely to do well with reduced intensity treatment first-line chemotherapy while preserving the current high efficacy, and how to augment therapy for patients with a high risk of relapse remains a critical yet unsatisfactory question. Thus, understanding the molecular mechanism(s) leading to chemo-resistance in relapsed patients may reveal possible novel drug targets and combination therapies to improve chemotherapy efficacy. Multiple treatment-resistance mechanisms will be identified, and it will be important to identify driver from passenger events among these mechanisms. More importantly, developing targeted therapies or precision medicine for each individual patient based on the unique genetic signatures from patients would be a pivotal direction to improve the treatment outcomes. This requires more in-depth investigations on both molecular mechanistic studies as well as pre-clinical and clinical examinations of newly proposed therapies. The development of PDX or PDOX models in immune-competent genetic murine models could facilitate this process.

## Figures and Tables

**Figure 1 cancers-15-00382-f001:**
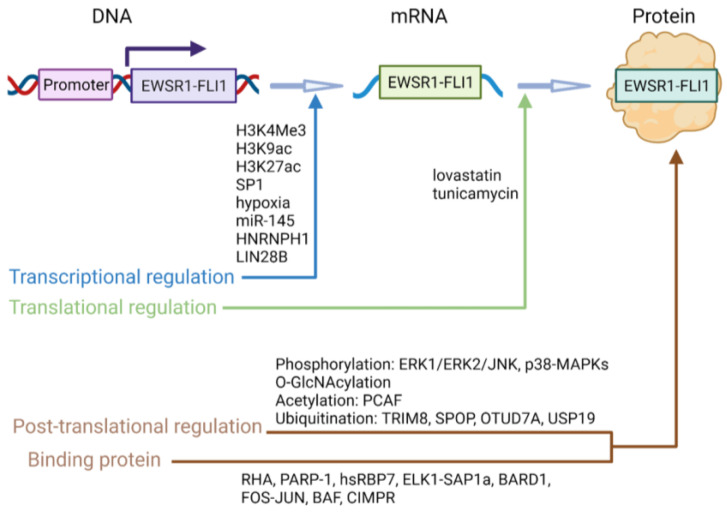
Regulatory mechanisms controlling EWSR1::FLI1. Proper EWSR1::FLI1 protein expression is controlled by multi-layer mechanisms, including regulation of transcription by epigenetic regulations, transcription factors and miRNA/lncRNAs, translational regulations, post-translational regulation and various binding proteins. This figure is generated using BioRender.

**Figure 2 cancers-15-00382-f002:**
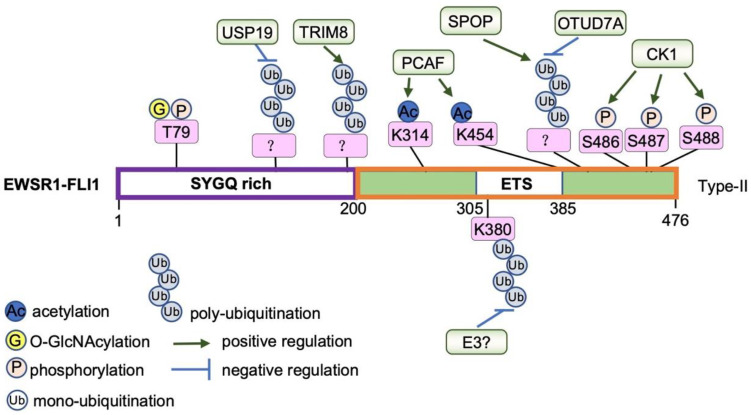
A summary of EWSR1-FLI1 post-translational modifications. Function of EWSR1-FLI1 proteins are regulated by various post-translational modifications including mono-ubiquitination, poly-ubiquitination, phosphorylation, acetylation and O-GlcNAcylation in cells as indicated here.

**Table 1 cancers-15-00382-t001:** A summary of FET–ETS fusion oncogenes in Ewing sarcoma.

**FET**	**ETS**	**Fusion Gene**	**Frequency**	**Translocation**
** *EWSR1* **	*FLI1*	*EWSR1-FLI1*	85%	t(11; 22)(q24; q12)
** *EWSR1* **	*ERG*	*EWSR1-ERG*	10%	t(21; 12)(q22; q12)
** *EWSR1* **	*FEV*	*EWSR1-FEV*	<1%	t(2; 22)(q33; q12)
** *EWSR1* **	*ETV1*	*EWSR1-ETV1*	<1%	t(7; 22)(p22; q12)
** *EWSR1* **	*E1AF*	*EWSR1-E1AF*	<1%	t(17; 22)(q21; q12
** *FUS* **	*FEV*	*FUS-FEV*	<1%	t(2; 16)(q35; p11)
** *FUS* **	*ERG*	*FUS-ERG*	<1%	t(16; 21)(p11; q22)
**ETS**	**FET**	**Fusion gene**	**Frequency**	**Translocation**
** *FLI1* **	*EWSR1*	*FLI1-EWSR1*	TBD	t(22; 11)(q12; q24)

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
