# Peer review of "Regulation of EWSR1-FLI1 Function by Post-Transcriptional and Post-Translational Modifications"

_cancers, 2023, doi:10.3390/cancers15020382_

Round 1
Reviewer 1 Report
Yu and his colleagues present a manuscript "Regulation of EWSR1-FLI1 function by post-transcriptional and post-translational modifications" reviewed the post-transcriptional and post-translational modifications of EWSR1-FLI1 protein in Ewing sarcoma. They carefully summarized the fusion oncogenes in this tumor. They also listed all regulatory network of EWSR1-FLI1 and the post-translational modifications. The summarizing data presented in the manuscript are highly convincing and the manuscript is very well-written. There are a few minor concerns that should be clarified prior to publication.
1. Discussion of the potential formation reason of these fusions, especially EWSR1-FLI1 caused by frequent translocation of chr11 and chr22.
2. Line 132 should add references.
Author Response
We deeply appreciate the constructive suggestions provided by the editor and reviewers during the initial review of our manuscript. The comments and suggests have been very helpful in guiding us to further improve our study. Following these suggestions, we have revised our manuscript. We hope that the editor and the reviewers agree that we have fully addressed all the reviewers’ concerns and substantially strengthened our paper. Therefore, we believe that the revised manuscript is now suitable for publication in Cancers.
Yu and his colleagues present a manuscript "Regulation of EWSR1-FLI1 function by post-transcriptional and post-translational modifications" reviewed the post-transcriptional and post-translational modifications of EWSR1-FLI1 protein in Ewing sarcoma. They carefully summarized the fusion oncogenes in this tumor. They also listed all regulatory network of EWSR1-FLI1 and the post-translational modifications. The summarizing data presented in the manuscript are highly convincing and the manuscript is very well-written. There are a few minor concerns that should be clarified prior to publication.
Response: We thank the reviewer for acknowledging our efforts in summarizing related EWSR1-FLI1 regulatory mechanisms to provide possible directions for treating Ewing sarcoma.
- Discussion of the potential formation reason of these fusions, especially EWSR1-FLI1 caused by frequent translocation of chr11 and chr22.
Response: We thank the reviewer for this great suggestion. We have included the statement for the formation of fusion gene in the discussion section in the revised manuscript.
- Line 132 should add references.
Response: We apologize for missing these references and have included them in the revised manuscript.

Reviewer 2 Report
Cancers, Lee Yu et al.,
Chapter 1, table 1
This describes the translocations. Please discus (brief) the possible reciprocal translocations.
Page 2, line 70
“The disease occurs with a slightly higher rate in male than female”. There’s also an ethnic bias. Please mention this.
Page 3, line 85-88
In addition to CD99 staining and sequencing, diagnosis using DNA methylation arrays is being tested. Please mention this.
Page 3, line 97
Here, survival percentages are discussed. 5 year survival has increased from 20% to 70%, but many patients relapse. In that case changes are poor. What is the survival percentage after 10 or 15 years
Page 3, line 111
LSD1 is mentioned as therapeutic target, Briefly mention what LSD1 is, this will make it more clear why this protein has been chosen as target, given the mechanism by which EWFSR1/FLI transforms cells.
Page 4, line 144
“MK-STYX (a MAPK 144 phosphatase-like protein) identified by this approach was further validated [45]”. What was the conclusion?
Page 4, line 152
Neuroblastmas should be neuroblastomas
Page 4, line 149 – 158
Here, the authors discuss a possible core-program from a meta-analysis of target genes in several cell (line) models. I’d like to see a critical appraisal of these data, especially in the light of earlier made remarks that the levels of EWSR1/FLI are critical. Given that many of the transcriptome data were obtained in immortal “cancer” cell lines or cell that might not be the elusive cell-of-origin, by expressing EWSR1/FLI at levels that are most likely not physiological.
Page 5, line 217 – 223
In the chapter on translational regulation as well as in figure 1, lovastatin and tunicamycin are mentioned as drugs that decrease EWSR1-FLI protein expression. This suggests the use of these compounds as potential therapy. In this light it would be good to refer to the study by Franzetti et al., Oncogene volume 36, pages 3505–3514 (2017), that indicates that lowering EWSR1/FLI levels lead to increased migration potential. This should be more emphasized throughout the manuscript. Lower EWSR1-FLI1 levels are not necessarily good.
Page 6, Figure 2
Please use different fonts/no bold for the domains in the protein.
Page 6, figure 2 and line 236 - chapter post-translational control
In this figure and chapter, post-translational modifications found in EWSR1-FLI, EWSR1 and FLI are discussed. Although briefly mentioned that it is currently unclear if the observed modifications are present in both the wt proteins and the fusion protein, the figure suggests that EWSR1-FLI has many unique modifications. This is misleading and might suggest that these modifications could be potential target sites for future therapies. If these modifications are also present in the wt proteins, there might be dramatic unwanted side-effects.
It would be better to show only the fusion protein and it’s modifications and emphasize that it’s currently unclear if these modifications are also present in the wt proteins (which is very likely).
Page 8, line 324 – further
In this part the lack of genetic mouse models is discussed. The authors mention that such a mouse model would shed light on the cell-of-origin and the first steps of oncogenesis. The latter is true, but the generation of a representative mouse model is a catch-22 situation since the number of combinations of unique cell types and developmental stages at which the EWSR1-FLI fusion gene needs to be expressed makes it almost impossible to test all of these. It would be better to rephrase this and say that a mouse model would confirm the cell-of-origin rather than reveal it.
Page 8, line 328 – 331
This sentence is not finished and it’s unclear to me what its relevance is.
Page 8, line 342 – 345
This part described the advantages of orthotopic vs flank transplantation in PDX models. Currently this part is confusing “in situ xenograft tumors do not metastasize whereas in situ transplanted tumors affect tumor metastasis”. Please rephrase this and directly use flank transplantation vs orthotopic transplantation.
In light of this, the publication by Stewart et al., Nature. 2017 Sep 7; 549(7670): 96–100 is worth mentioning.
Author Response
We deeply appreciate the constructive suggestions provided by the editor and reviewers during the initial review of our manuscript. The comments and suggests have been very helpful in guiding us to further improve our study. Following these suggestions, we have revised our manuscript. We hope that the editor and the reviewers agree that we have fully addressed all the reviewers’ concerns and substantially strengthened our paper. Therefore, we believe that the revised manuscript is now suitable for publication in Cancers.
Reviewer #2
Chapter 1, table 1
This describes the translocations. Please discus (brief) the possible reciprocal translocations.
Response: We thank the reviewer for this great suggestion and have included this reciprocal translocation in Table 1 with a brief discussion in main text.
Page 2, line 70
“The disease occurs with a slightly higher rate in male than female”. There’s also an ethnic bias. Please mention this.
Response: We thank the reviewer for raising this question. Since this statement is not critical and essential to the review with less support from research data, we have removed this sentence from the revised manuscript to avoid any further confusions from readers.
Page 3, line 85-88
In addition to CD99 staining and sequencing, diagnosis using DNA methylation arrays is being tested. Please mention this.
Response: We thank the reviewer for this great suggestion. Following the reviewer’s suggestion, we have included DNA methylation tests in the revised manuscript.
Page 3, line 97
Here, survival percentages are discussed. 5 year survival has increased from 20% to 70%, but many patients relapse. In that case changes are poor. What is the survival percentage after 10 or 15 years
Response: We thank the reviewer for this great suggestion. We have included that the survival rate for relapsed patients is less than 30% in the revised manuscript.
Page 3, line 111
LSD1 is mentioned as therapeutic target, Briefly mention what LSD1 is, this will make it more clear why this protein has been chosen as target, given the mechanism by which EWFSR1/FLI transforms cells.
Response: Following the reviewer’s suggestion, we have included statements for LSD1 function and recruitment by EWSR1-FLI1 in Ewing sarcoma to suppress key EWSR1-FLI1 targets in the revised manuscript.
Page 4, line 144
“MK-STYX (a MAPK 144 phosphatase-like protein) identified by this approach was further validated [45]”. What was the conclusion?
Response: Following the reviewer’s suggestion, we have completed this sentence in the revised manuscript.
Page 4, line 152
Neuroblastmas should be neuroblastomas
Response: We apologize for this typo and have corrected it in the revised manuscript.
Page 4, line 149 – 158
Here, the authors discuss a possible core-program from a meta-analysis of target genes in several cell (line) models. I’d like to see a critical appraisal of these data, especially in the light of earlier made remarks that the levels of EWSR1/FLI are critical. Given that many of the transcriptome data were obtained in immortal “cancer” cell lines or cell that might not be the elusive cell-of-origin, by expressing EWSR1/FLI at levels that are most likely not physiological.
Response: We fully agree with the reviewer on this excellent suggestion and have included a statement as suggested in the revised manuscript.
Page 5, line 217 – 223
In the chapter on translational regulation as well as in figure 1, lovastatin and tunicamycin are mentioned as drugs that decrease EWSR1-FLI protein expression. This suggests the use of these compounds as potential therapy. In this light it would be good to refer to the study by Franzetti et al., Oncogene volume 36, pages 3505–3514 (2017), that indicates that lowering EWSR1/FLI levels lead to increased migration potential. This should be more emphasized throughout the manuscript. Lower EWSR1-FLI1 levels are not necessarily good.
Response: We thank the reviewer for this great suggestion and fully agree with reviewer that we should emphasize that lower EWSR1-FLI1 protein levels may facilitate cell migration. We have included this statement as well as the suggested reference in the revised manuscript.
Page 6, Figure 2
Please use different fonts/no bold for the domains in the protein.
Response: We apologize for this hard-reading fonts which might be generated during format conversions. We have revised the fonts to make them more standing out and easier for reading in the revised Figure 2.
Page 6, figure 2 and line 236 - chapter post-translational control
In this figure and chapter, post-translational modifications found in EWSR1-FLI, EWSR1 and FLI are discussed. Although briefly mentioned that it is currently unclear if the observed modifications are present in both the wt proteins and the fusion protein, the figure suggests that EWSR1-FLI has many unique modifications. This is misleading and might suggest that these modifications could be potential target sites for future therapies. If these modifications are also present in the wt proteins, there might be dramatic unwanted side-effects.
It would be better to show only the fusion protein and it’s modifications and emphasize that it’s currently unclear if these modifications are also present in the wt proteins (which is very likely).
Response: We thank the reviewer for this great suggestion and fully agree with the reviewer that it is unknown if the modifications occurring on EWSR1-FLI1 fusion proteins are also observed in WT proteins. Following the reviewer’s suggestion, we have removed all EWSR1 and FLI1 modifications and revised Figure 2 as well.
Page 8, line 324 – further
In this part the lack of genetic mouse models is discussed. The authors mention that such a mouse model would shed light on the cell-of-origin and the first steps of oncogenesis. The latter is true, but the generation of a representative mouse model is a catch-22 situation since the number of combinations of unique cell types and developmental stages at which the EWSR1-FLI fusion gene needs to be expressed makes it almost impossible to test all of these. It would be better to rephrase this and say that a mouse model would confirm the cell-of-origin rather than reveal it.
Response: We thank the reviewer for this great suggestion and fully agree with the reviewer. Following the reviewer’s suggestion, we have changed reveal to confirm in the revised manuscript.
Page 8, line 328 – 331
This sentence is not finished and it’s unclear to me what its relevance is.
Response: We apologize for missing the last part of this sentence. In the revised manuscript we have completed this sentence.
Page 8, line 342 – 345
This part described the advantages of orthotopic vs flank transplantation in PDX models. Currently this part is confusing “in situ xenograft tumors do not metastasize whereas in situ transplanted tumors affect tumor metastasis”. Please rephrase this and directly use flank transplantation vs orthotopic transplantation.
In light of this, the publication by Stewart et al., Nature. 2017 Sep 7; 549(7670): 96–100 is worth mentioning.
Response: We thank the reviewer for raising this great suggestion. Following the reviewer’s suggestion, we have rephrased PDX and PDOX and included the indicated reference.

Reviewer 3 Report
Here, Yu and colleagues provide a review article detailing how EWSR1-FLI1 function is Ewing sarcoma cells is regulated through both post-transcriptional and post-translational modifications. Ewing sarcoma is a devastating pediatric bone cancer and approximately 85% of patients will harbor the EWSR1-FLI1 fusion oncoprotein in their tumor cells. The ubiquity of the fusion oncoprotein and its strength as a driving mutation in Ewing sarcoma cells necessitates the compilation of this review article as the authors due an admirable job in assimilating what is known in the literature about EWSR1-FLI1 regulation. The references are chosen well and assembled in a mostly clear, concise manner that requires minimal editing for more pristine clarity. My comments are listed below.
1) Please double-check gene and long non-coding RNA nomenclature and the use of EWSR1-FLI1/EWS-FLI1 throughout the manuscript.
a. For example, “EWSAT1” (line 161) should be italicized.
b. The use of EWS-FLI1 and EWSR1-FLI1 is frequently interchanged throughout the manuscript. Please define the reason why these two names are interchanged or choose one for the clarity of the reader.
2) The cell of origin for human Ewing sarcomas is still debated and other sources aside from mesenchymal stem cells have been suggested. Please include in line 59 and 60 of the Introduction other possible cells of origin including neural stem cells (von Levetzow et al (2011) Plos One) or osteochondrogenic progenitors (Tanaka et al (2014) J Clin Invest).
3) There are a couple of instances (line 132 and line 214) where references need to be added. Line 132 says to add references. Line 214 has PMIDs. Please add references accordingly.
4) In line 154, the authors list FOXOM1 as a target of EWSR1-FLI1. Do the author’s mean FOXO1 or FOXM1? There is no FOXOM1 that the reviewer is aware.
5) There are discrepancies in how the subheadings are presented in sections 5.1 and 5.3. 5.1 has no special fonts, but in 5.3 each paragraph subheading is bolded. Please make uniform for the clarity of the reader.
6) In line 349, the term “magic powers” seems out of place in the manuscript for describing the efficacy of immunotherapies. Please revise to illustrate that immune therapies are powerful therapeutic options for cancer treatment in this section of the manuscript.
7) In Figure 2, the wording in the boxes for the different domains of EWSR1-FLI1 are shaded and make it hard to read for the audience. Please modify the domain names to be more legible. The other boxes and circles are fine.
Author Response
We deeply appreciate the constructive suggestions provided by the editor and reviewers during the initial review of our manuscript. The comments and suggests have been very helpful in guiding us to further improve our study. Following these suggestions, we have revised our manuscript. We hope that the editor and the reviewers agree that we have fully addressed all the reviewers’ concerns and substantially strengthened our paper. Therefore, we believe that the revised manuscript is now suitable for publication in Cancers.
Reviewer #3
Here, Yu and colleagues provide a review article detailing how EWSR1-FLI1 function is Ewing sarcoma cells is regulated through both post-transcriptional and post-translational modifications. Ewing sarcoma is a devastating pediatric bone cancer and approximately 85% of patients will harbor the EWSR1-FLI1 fusion oncoprotein in their tumor cells. The ubiquity of the fusion oncoprotein and its strength as a driving mutation in Ewing sarcoma cells necessitates the compilation of this review article as the authors due an admirable job in assimilating what is known in the literature about EWSR1-FLI1 regulation. The references are chosen well and assembled in a mostly clear, concise manner that requires minimal editing for more pristine clarity. My comments are listed below.
1) Please double-check gene and long non-coding RNA nomenclature and the use of EWSR1-FLI1/EWS-FLI1 throughout the manuscript.
- For example, “EWSAT1” (line 161) should be italicized.
Response: We thank the reviewer for these great suggestions. We didn’t italicize genes/RNAs in the original manuscript because we thought the journal would format these prior to publication during the production phase. Following the reviewer’s suggestion, we have italicized genes/RNAs in the revised manuscript, including genes in Table 1.
- The use of EWS-FLI1 and EWSR1-FLI1 is frequently interchanged throughout the manuscript. Please define the reason why these two names are interchanged or choose one for the clarity of the reader.
Response: We thank the reviewer for raising this concern and apologize for the inconsistent uses of EWSR1-FLI1 across the original manuscript. EWS-FLI1 and EWSR1-FLI1 are referring to the same fusion gene. To avoid future confusions, we have replaced all EWS-FLI1 with EWSR1-FLI1 in the revised manuscript.
2) The cell of origin for human Ewing sarcomas is still debated and other sources aside from mesenchymal stem cells have been suggested. Please include in line 59 and 60 of the Introduction other possible cells of origin including neural stem cells (von Levetzow et al (2011) Plos One) or osteochondrogenic progenitors (Tanaka et al (2014) J Clin Invest).
Response: We thank the reviewer for raising this great suggestion and we have included these two references in the revised manuscript to indicate other possible origins.
3) There are a couple of instances (line 132 and line 214) where references need to be added. Line 132 says to add references. Line 214 has PMIDs. Please add references accordingly.
Response: We apologize for missing these references and have included them in the revised manuscript.
4) In line 154, the authors list FOXOM1 as a target of EWSR1-FLI1. Do the author’s mean FOXO1 or FOXM1? There is no FOXOM1 that the reviewer is aware.
Response: We apologize for this typo- it should be FOXM1 and we have corrected it in the revised manuscript. We have also carefully read through the manuscript again to eliminate any spotted typos.
5) There are discrepancies in how the subheadings are presented in sections 5.1 and 5.3. 5.1 has no special fonts, but in 5.3 each paragraph subheading is bolded. Please make uniform for the clarity of the reader.
Response: We have removed bolded fonts format in 5.3. section to be consistent with 5.1. in the revised manuscript.
6) In line 349, the term “magic powers” seems out of place in the manuscript for describe the efficacy of immunotherapies. Please revise to illustrate that immune therapies are powerful therapeutic options for cancer treatment in this section of the manuscript.
Response: We thank the reviewer for raising this suggestion and have made corresponding changes in the revised manuscript.
7) In Figure 2, the wording in the boxes for the different domains of EWSR1-FLI1 are shaded and make it hard to read for the audience. Please modify the domain names to be more legible. The other boxes and circles are fine.
Response: We apologize for this hard-reading fonts which might be generated during format conversions. We have revised the fonts to make them more standing out and easier for reading in the revised Figure 2.
